# Analyzing Patterns of Service Utilization Using Graph Topology to Understand the Dynamic of the Engagement of Patients with Complex Problems with Health Services

Jonas Bambi [1], Yudi Santoso [2], Ken Moselle [3], Stan Robertson [4], Abraham Rudnick [5,*], Ernie Chang [6] and Alex Kuo [1]

1   Department of Health Information Science, Faculties of Human and Social Development, Victoria Campus, University of Victoria, Victoria, BC V8P 5C2, Canada; jonasbambi@uvic.ca (J.B.); akuo@uvic.ca (A.K.)
2   Independent Researcher, Victoria, BC V8R 5B4, Canada; y.santoso8@gmail.com
3   Department of Clinical Psychology, Faculty of Social Science, Victoria Campus, University of Victoria, Victoria, BC V8P 5C2, Canada; kmoselle@uvic.ca
4   Independent Researcher, Victoria, BC V8Y 2W3, Canada; stanrobertson@shaw.ca
5   Departments of Psychiatry and Bioethics, School of Occupational Therapy, Faculties of Medicine and Health, Dalhousie University, Halifax, NS B3H 4R2, Canada
6   Retired Physician, and Independent Computer Scientist, Victoria, BC V9C 4B1, Canada; ecsendmail@gmail.com
*   Correspondence: abraham.rudnick@nshealth.ca

**Abstract:** Background: Providing care to persons with complex problems is inherently difficult due to several factors, including the impacts of proximal determinants of health, treatment response, the natural emergence of comorbidities, and service system capacity to provide timely required services. Providing visibility into the dynamics of patients' engagement can help to optimize care for patients with complex problems. Method: In a previous work, graph machine learning and NLP methods were used to model the products of service system dynamics as atemporal entities, using a data model that collapsed patient encounter events across time. In this paper, the order of events is put back into the data model to provide topological depictions of the dynamics that are embodied in patients' movement across a complex healthcare system. Result: The results show that directed graphs are well suited to the task of depicting the way that the diverse components of the system are functionally coupled—or remain disconnected—by patient journeys. Conclusion: By setting the resolution on the graph topology visualization, important characteristics can be highlighted, including highly prevalent repeating sequences of service events readily interpretable by clinical subject matter experts. Moreover, this methodology provides a first step in addressing the challenge of locating potential operational problems for patients with complex issues engaging with a complex healthcare service system.

**Keywords:** clinical pathways; clinical practice guideline; decision support; electronic healthcare; graph topology; health information management; health service system; machine learning algorithms; hub and spoke; patients' engagement dynamics

## 1. Introduction

### 1.1. Engagement of Patients with the Healthcare Service System

Persons and populations contending with chronic and progressively more complex chronic health conditions will typically interact with an increasingly diverse array of health services over time [1,2]. The clinical trajectories for such persons, and the experience of sentinel clinical events, such as organ failure or the loss of functional capacity, are an emergent characteristic of the dynamics of persons interacting with a complexly structured health service system.

Providing care to patients with complex problems, while relying solely on the traditional taxonomical diagnostic approach to care management, can be a challenge for the healthcare system and may not be of benefit to the patients [3]. This approach is also inherently limited. Chronic disease treatment and management consume over 42% of the total direct medical care expenditure in Canada [4], over 50% in the US [5,6], and is a key factor driving the overall growth in spending in the traditional Medicare program [7]. Most notably, outcomes will be impacted by the dynamics of patients with complexly emerging comorbidities interacting with a changing array of services that respond to these emerging problems and diagnoses. In this case, outcomes are an emergent characteristic of the dynamic interplay of the pathophysiology of chronic diseases, the proximal determinants of the health profile of the patient (e.g., health risk behavior; adherence to treatment protocols), and the accessibility of services, which itself is complexly determined by funding, geography, and public health emergencies that compete for resources, just to name a few.

In [8,9], graph machine learning (ML) and natural language processing (NLP) methods were used to model the products of service system dynamics as atemporal entities, using a data model that collapsed patient encounter events across time. The result is clusters of services where the clustering reflects functional proximity and the co-emergence of access to multiple services within individual patient records. The resulting static models show how closely positioned services are to one another. However, this model failed to capture the strength of the coupling between services as well as the order of service access events. One can optimize service system operations from such models when the outcomes for patients are a simple linear combination of the services or impacts or effects of each component. In this case, optimization is a direct extrapolation from the population incidence/prevalence of focal conditions. The structures and functional organization of the components of the service system map cleanly onto discrete diagnostic entities.

To relate outcomes to the dynamics of patients interacting with different components of the service system over time, the order of events needs to be incorporated back into the models created in [8,9]. The objective is to provide visibility into the interoperation of the multiple components of the service system based on the prevalent longitudinal features of patient journeys through the health service system. To the extent that outcomes for a complex chronic patient level are coupled with those dynamics, these models of service system dynamics are essential for service system optimization.

This paper presents a methodology for discovering and depicting the dynamics that characterize the interoperation of multiple components of a complex health service system, in relation to cohorts contending with increasingly impactful chronic conditions. These dynamics appear as longitudinal patterns of service utilization, sparsely distributed in a high-dimensional array of services spanning a full array of secondary and tertiary services. Secondary and tertiary services include hospital and community-based services, outreach, residential care, case management, and numerous others, for medical/surgical problems such as cardiovascular disease as well as mental health and addiction services.

### 1.2. Graph/Network Modeling and Analysis

Since the last century, network/graph analysis has become an indispensable tool for analyzing systems whose structures and dynamics are embodied in patterns of discrete interactions among large arrays of elements [10,11]. The network models of service system dynamics provide a basis for designing and staging interventions that will alter those dynamics and associated outcomes. This is compared to interventions that increase the supply of select components in a system, without altering the dynamics. More of the same dynamics can be expected to engender more of the same outcomes.

What links together a very large body of work using graph analysis for a very diverse array of problems is a base representation of the source data as a set of entities (nodes) and connections between the nodes (edges) that reflect the dynamics of the system. Those edges may reflect asymmetrical processes where something associated with one node either precedes some other nodal event in time, or there is some form of causal relationship

between one node and another. An example would be pathways through the service system that originate in one location, e.g., an emergency department, and connect at some future point in time to another service, e.g., an electro-diagnostic procedure, or admission to an acute care bed, or a residential care facility, or a morgue encounter. These nodes and edges form directed graphs.

As well, the edges may reflect symmetrical relationships among events that may not be directly related but are instead linked to some common factor. An example would be two health services that are not functionally connected but are linked by patients who access both. In healthcare, the mediating or connecting function may arise from the patients, not the service protocols that are embodied by programs in the system. In such a case, the nodes and edges will form an undirected graph.

Both undirected and directed graph models and analytics have a role in understanding the clinical dynamics of patients and the service system dynamics that emerge as patients interact over time with the service system. If one is concerned with the ways in which the components of the service system are related to one another, one may employ a static representation of the service system represented by an undirected graph. That is what was illustrated in [8,9]. In that case, the goal is to know how functionally proximal (and therefore accessible) are different services from one another. For example, how "far" is a withdrawal management service from a post-withdrawal stabilization bed vs a residential care facility for persons contending with severe psychiatric illness, such as schizophrenia.

If one is concerned with how the clinical/functional/behavioral risk profile of patients (over time) impact the service system operations, performing graph analytics within some clinically characterizable cohorts may be needed. An example would be a cohort of persons contending with a severe psychiatric illness with a comorbid substance use disorder, or persons contending with a chronic-relapsing condition that is not associated with an underlying severe psychiatric condition. The components of the service system may interoperate differently for these two cohorts. If the concern is with dynamics, the underlying representation of the data may consist of a directed graph.

In healthcare, network analysis has been used in many areas: (1) in precision medicine, by linking intracellular structures and processes to diagnostically relevant features [12]; (2) in the extraction and interpretation of clinically relevant signs and symptoms from a large array of sources relating to a diverse array of diagnostic entities [13]; (3) in the understanding of disease–gene associations, with a focus on understanding the construction of human disease network, diseasome [14]; (4) in the area of precision diagnostics and the facilitation in accelerating the diagnosis of rare or previously unrecognized diseases [12]; (5) in supporting various predictions, including diagnosis, patient clinical outcomes, and readmission [15]; (6) in improving healthcare quality through clinical practice guideline, consisting of protocols relying on diverse bodies of clinical information and treatments provided [16,17], and depicted as cohort-specific recurring patterns of service utilization that actually take place within a network of local services—service pathways [18]; and (7) in the depiction of patient journeys, assembled from one or more service pathways, in response to patients contending with possibly multiple co-occurring or emerging problems [19].

There is a very large body of works that cover the first six areas referenced above. However, the literature becomes far sparser when it comes to the seventh area, which is concerned with the *de facto* clustering of services based on the interaction of the cross-continuum system with diverse clinical populations. To construct these cross-continuum longitudinal models, the source data must have a good coverage of the full space of services, so that key services contributing to the dynamics are contained within the set of events out of which the models are constructed. Because patient journeys in these high-dimensional spaces are almost invariably distinguishable, the source data carries privacy risks due to risk of re-identification [20]. This militates against the public release of the datasets required to construct these models, and that is a likely contributor to the sparseness of the published results.

The work set out in this paper was undertaken by a team located inside the firewalled boundaries of a regional health authority. This team had access to a complete patient journey dataset, de-identified to the point where the risk of identity disclosure is managed, but without the perturbative changes to the core clinical contents that would be required to render the source data in a form that would, at least, arguably be suitable for public release—see, for example, the differential privacy for one such approach [21,22].

Using these data, in this paper, a method is proposed for viewing longitudinal healthcare encounter data, spanning a cross-continuum service system, as a directed network, to uncover some network topologies, which highlight the way services are in effect coupled by patients as they access the services over time. The objective is to understand the dynamic of the engagement of patients with complex problems with the healthcare services. Therefore, the analogy of physical network topology [23] to highlight these dynamic features is used.

## 2. Objectives

The focus of this study was on revealing the dynamics of service system interoperations, reflecting the movement of patients with complex issues through the service system. Revealing patient dynamics from a cross-continuum complex healthcare dataset in such a way that their depictions are clear in informing the effort to change patients' outcome by altering those dynamics is challenging. Hence, to address this, the following questions are answered in this work:

- Can patient journeys be recorded as sequences of service encounters, using a directed graph that can then be used to identify high-prevalence sequences within and across persons?
- With the proposed methodology, to what extent can one cut across the complexity of a cross-continuum service structure to capture the dynamics of the journey of patients with complex issues that clearly portray their engagement with the service system, to help to locate potential operational problems?

## 3. Methods

### 3.1. Addressing Data Granularity Issues

The health organization whose data were being used for this study provides a comprehensive array of secondary and tertiary health services. This includes acute care/intensive care services, hospital and community-based emergency response, ambulatory services, residential care services for older adults or persons contending with mental health issues, case management services, and a range of addiction harm reduction or rehab and recovery-oriented services. A certificate of approval was provided by the Research Ethics Board to conduct this research project, under the study number H21-02817.

One or more of these services are encapsulated into an array of roughly 2000 Service Units within a location built for a clinical information system used to support the delivery of care. Service unit names within the local clinical information system are often opaque, rendering them unsuitable for supervised machine learning methods that require meaningfully labeled data. As well, the service units may vary widely with regard to granularity—for example, multiple beds will appear as a single unit within an acute care facility, but multiple beds in a large array of family care homes for frail elderly will show up as multiple service units.

To address these issues, a clinical context coding scheme (CCCS) was developed [24]. This scheme is organized around six sets of codes, constituting a semantic layer applied to all of the 2000 Service Units. The roughly 200 Service Classes employed for the modeling in this paper consist of equivalence classes formed by the application of these code sets to those service units.

### 3.2. Representing Patient Access to Healthcare Service Systems as a Graph

In order to study the dynamics of the healthcare system, service utilization is visualized as a directed graph through the following method. In the encounter dataset, there are

patients and records of their interactions with service classes. One patient typically uses several service classes throughout his/her journey. A given patient may also access a given service class on more than one occasion, e.g., repeat admissions to an Emergency Department or routine repeated blood work. If the records of a patient are arranged sequentially in chronological order, one service can be connected to another by a "next service" relationship. Suppose patient A used service class 1, and then 2, and then 2 again, and then 3, and then 2. This sequence can be depicted as a directed graph with the service classes as nodes and directed edges that reflect the transitions. This is illustrated in Figure 1.

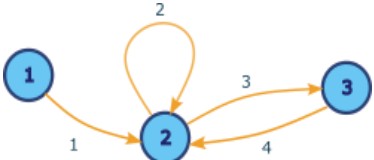

**Figure 1.** A directed graph of three nodes and four edges, depicting a patient journey from service class 1 to 2, to 2 again, to 3 and back to 2. Here, the edge label is for the sequence order.

For each patient, a directed graph can be drawn based on their healthcare journey. Over the course of successive service events, the graph becomes more complex, with more nodes associated with services, and more edges connecting nodes by virtue of their co-occurrence within the longitudinal record of the patient.

If a set of patients or a cohort, rather than just a single patient, is considered, the directed graphs can be aggregated to generate a single-weighted directed graph. The weights of the edges can be defined through several formulations. First, the total number of transition instances can be used as the weight, which can be referred to as the raw weight. Second, the total number of patients within the cohorts that have ever undergone such transition can be used as the weight. A third method is to generate and use transition probabilities for each node, either 'to' or 'from'.

For a node x as an example, 'probability-from' can be viewed as the empirical transition probability from x for each edge, defined by the raw weight, divided by the total number of transitions from x—$Tf(x)$. Similarly, 'probability-to' can be computed by gathering the raw weight for each edges 'to' x divided by the total number of transitions to x—$Tt(x)$. This is illustrated in Figure 2. In Figure 2a, there is a partial view of the graph centering on x, with raw edge weight. With respect to x, there is $Tf(x) = 11$ and $Tt(x) = 25$. In Figure 2b, the edge weight is replaced by 'probability-to', while in Figure 2c, the edge weight is replaced by 'probability-from'. To compute the new weights for all edges in the graph, all nodes need to be considered one by one.

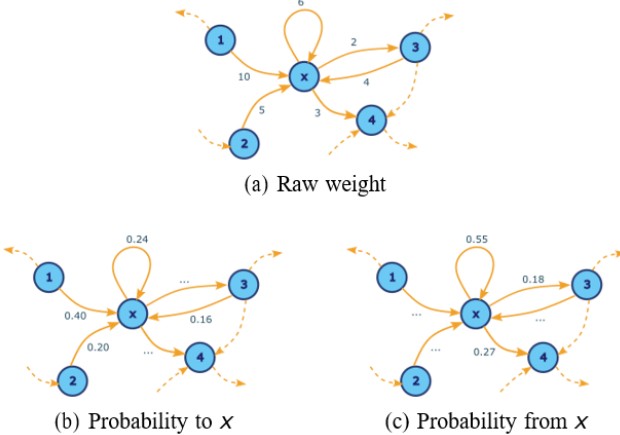

(a) Raw weight

(b) Probability to *x*        (c) Probability from *x*

**Figure 2.** An example of switching from raw weight to probabilistic weight: (**a**) represents a partial view of the graph centering on x, with raw edge weight, (**b**) the edge weight is replaced by 'probability-to', and (**c**) the edge weight is replaced by 'probability-from'.

If there are many patients in the cohort, the aggregated graph can be very dense, with the number of edges approaching the square of the number of nodes. It would be difficult to digest such a graph visually, especially when the number of nodes is also quite large. Therefore, there is a need to take a further step to illuminate the core topology of the graph.

Transitions that are of low prevalence are less likely to reflect features that are shared by the members of the cohort. As a result, (a) one may not know when that edge appears, and (b) one may not know whether one can extrapolate from that edge to another sample of persons who share the same cohort-defining properties. Hence, if one wants to extrapolate from this particular sample of persons with feature X, the selection of the most prevalent edges is needed. In such a case, an edge with a large weight is more important than an edge with a small weight. To see the features of service system dynamics that reflect the common characteristics of the cohort members, a filter for the highest prevalent next-service-event edges is applied to the graph, by limiting the number of edges shown. This number is set to a value such that the edges that reflect only a variation in the features of the cohort members and are not directly associated with the characteristics that define the cohort are filtered out. Clinical subject matter experts may be required to make that determination.

Appendix A provides additional details for the tools and algorithms used to generate the visualizations outlined in this study. This provides an in-depth technical understanding of the methodology used herein. It also makes it easier to implement approaches similar to the ones outlined herein to analyze complex healthcare dataset to generate topologies representing the dynamics of patients' engagements.

## 4. Analysis and Results

### 4.1. Cohort Selection, Analysis Setup, and Visualization Adjustment

For the analysis, the encounter data from one of the regional Health Authorities within Canada, covering the period from 2016 to early 2023, were used. Three cohorts were selected. These cohorts are quite diverse with regards to the underlying pathophysiologies. The proximal determinants of health that are intrinsic to people and their health conditions, including the pathophysiology of diseases and reasonably predictable clusters of co-emergent conditions, were some of the characteristics that were used for selecting the cohorts of interest. Hence, Cohort A had 2008 patients with a cohort definition based on diagnosis. Cohort B had 5397 patients, reflecting a different diagnostic profile. Cohort C had 15,063 patients, where the cohort-defining feature was a combination of clinical and demographic features. For the purpose of the work presented in this paper, it was not necessary to supply clinical details.

Figure 3a depicts the graph with a full set of edges for Cohort A. In Figure 3b,c, we see comparable graphs for Cohort B and Cohort C, with a full set of edges. At this level of resolution, it is difficult (at least visually) to detect any features of the graph topologies that would distinguish the products based on the three cohorts.

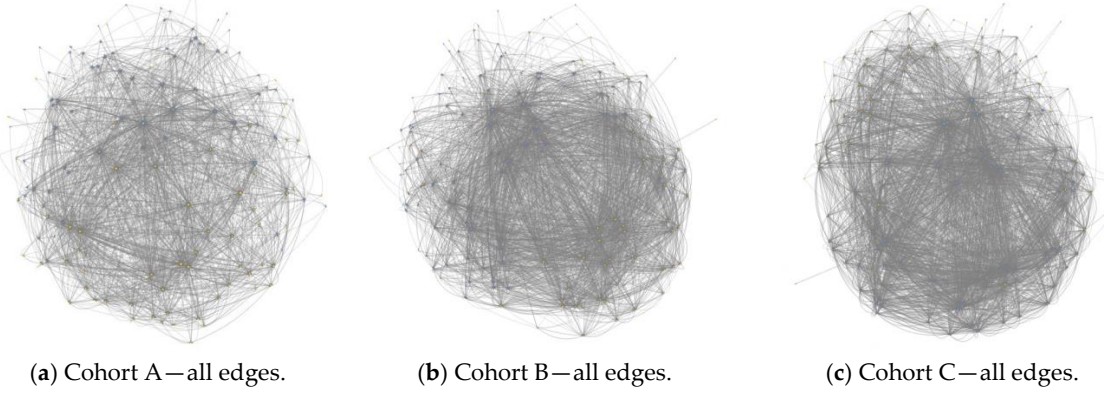

(**a**) Cohort A—all edges.      (**b**) Cohort B—all edges.      (**c**) Cohort C—all edges.

**Figure 3.** Comparing the visualizations for all three cohorts with full sets of edges: (**a**) represents cohort A, (**b**) represents cohort B, and (**c**) represents cohort C.

Focusing on Cohort A, moving from the picture of the graph with all edges (in Figure 4a) to the picture with 100 largest raw weight edges (in Figure 4b) and 50 edges (in Figure 4c), we see that the picture becomes clearer and clearer as the number of edges decreases. However, it should be noted, as previously mentioned, that the tuning for the most suitable number of edges will need to be determined by a clinical subject matter expert (SME) and/or a healthcare service system operation expert (SSOE)—i.e., those parties that have excellent cross-continuum knowledge of the mechanisms involved in navigating across service units.

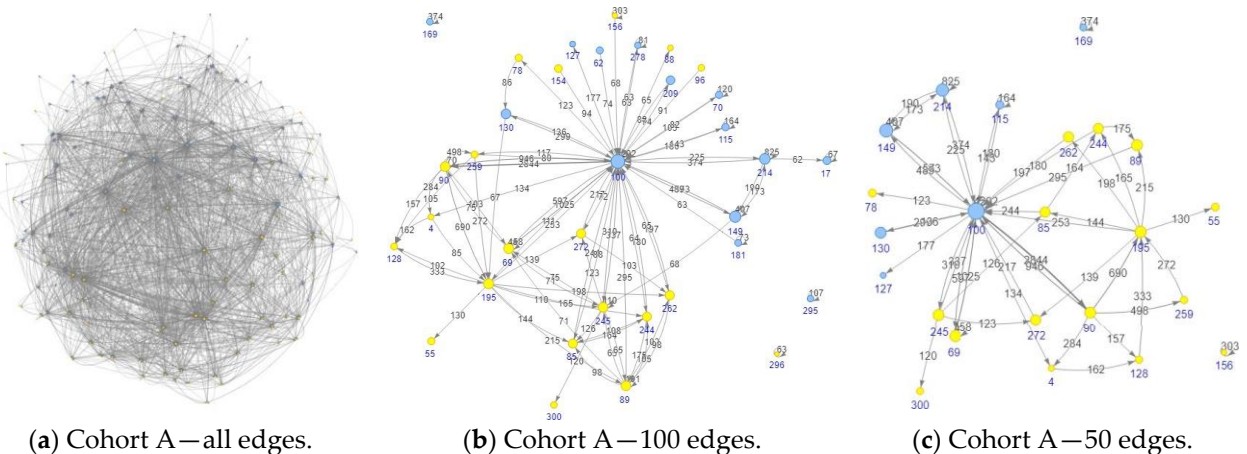

(**a**) Cohort A—all edges.     (**b**) Cohort A—100 edges.     (**c**) Cohort A—50 edges.

**Figure 4.** Demonstrates the need to adjust the number of the largest weight edges included in the visualization: (**a**) represents cohort A with all edges, (**b**) represents cohort A with 100 largest raw weight edges, and (**c**) represents cohort A with 50 largest raw weight edges.

A patient may contribute more than once to an edge weight. If the concern is only on the number of patients in each transition, the weight can be changed from raw weight to the number of patients. As the numbers of edges shown are dialed down, the apparent topology of the graph might look slightly different. The graph for Cohort A with fifty edges, using the number of patients, is shown in Figure 5. This is to be compared to Figure 4c.

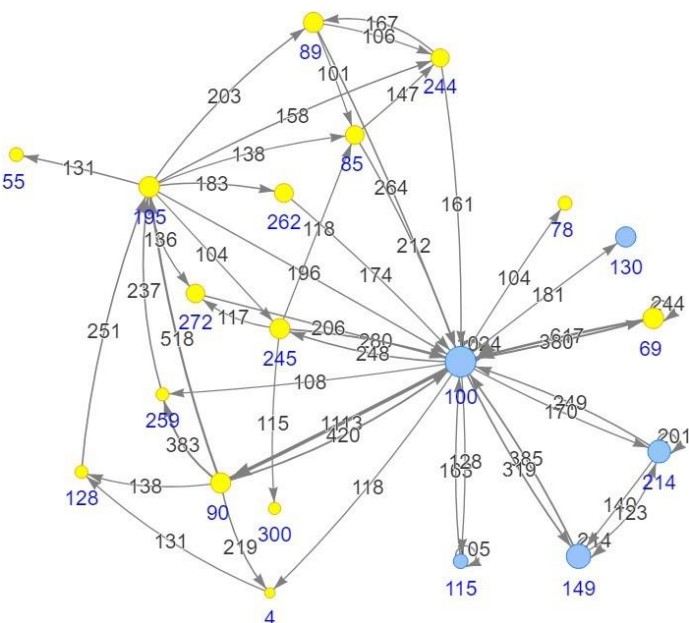

**Figure 5.** The graph for Cohort A using the number of patients as the edge weight and including only the 50 largest weight edges. The yellow nodes are for MHSU (mental health substance use)-related service classes, while the blue nodes are for non-MHSU or medical/surgical service classes.

*4.2. Cohort Topologies' Comparison*

Each node represents a service class. De-identified numeric IDs were used to simplify the picture. The yellow nodes are for MHSU (mental health substance use)-related service classes, while the blue nodes are for non-MHSU or medical/surgical service classes. What is striking for Cohort A, represented in Figure 6a, is that there is a close-knitted network among the MHSU service classes, while the non-MHSU service classes are only mainly connected to node 100. It can be noted that node 100 plays a central role in this graph. This happens to be the case for the other cohorts as well. By studying this plot, one can obtain some insights on how the service classes inter-operate for this cohort. For example, it can be noticed that many patients move from 90 to 195, either directly or through 128 or 259. Then, they move from 195 to some other MHSU service classes.

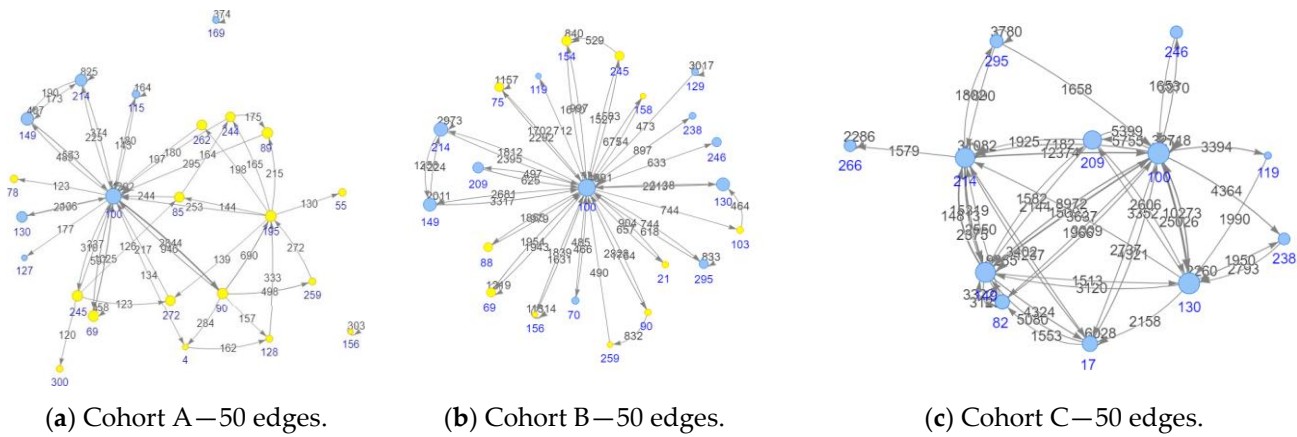

(**a**) Cohort A—50 edges.    (**b**) Cohort B—50 edges.    (**c**) Cohort C—50 edges.

**Figure 6.** Comparing the core topologies among the three cohorts: (**a**) cohort A, (**b**) cohort B, and (**c**) cohort C, all with 50 edges. The yellow nodes are for MHSU (mental health substance use)-related service classes, while the blue nodes are for non-MHSU or medical/surgical service classes.

Next, Cohort A was compared to the other two cohorts, Cohort B and Cohort C. For this comparison, the number of edges shown was fixed to 50. Note that the optimal number of edges to be shown in each cohort may differ from each other. However, for such diverse cohorts as the ones represented here, the differences in the topologies can still be seen without tuning to the most optimal value. Figure 6 is used to show the plots for these cohorts.

Here, the differences between the topologies of Cohort A and the Cohort B can be clearly seen. While in Cohort A there is a closely connected network among the many MHSU service classes, in Cohort B, the MHSU service classes are largely un-connected to each other, but rather, only connected to node 100. This forms a topology that can be referred to as a hub and spoke topology [25].

In Cohort C, the MHSU service classes are minor, used by only a small subset of patients. For this reason, they are not seen in Figure 6c, which includes only the 50 largest weight edges. Notably, the non-MHSU service classes (i.e., the medical/surgical service classes) form a closely connected network in this cohort. The node 214 and node 149 play strong roles in this network, followed by 130 and 17.

*4.3. Representing a Cohort Topology Using Transition Probability*

So far in this work, the number of instances and the number of patients have been used as the edge weight. Using other weights provides a way to look at the network from different perspectives. Focusing on Cohort A, the transition probabilities are now used as the weight. Figure 7 shows two plots, one with probability-from-edge-weight and another with probability-to-edge-weight, both with one hundred edges shown. They look different from those in Figure 4b, which uses the number of instances as the edge weight. Note that

these probabilistic weights are local measures, i.e., they are node-centered. Thus, they are seen from a node point of view.

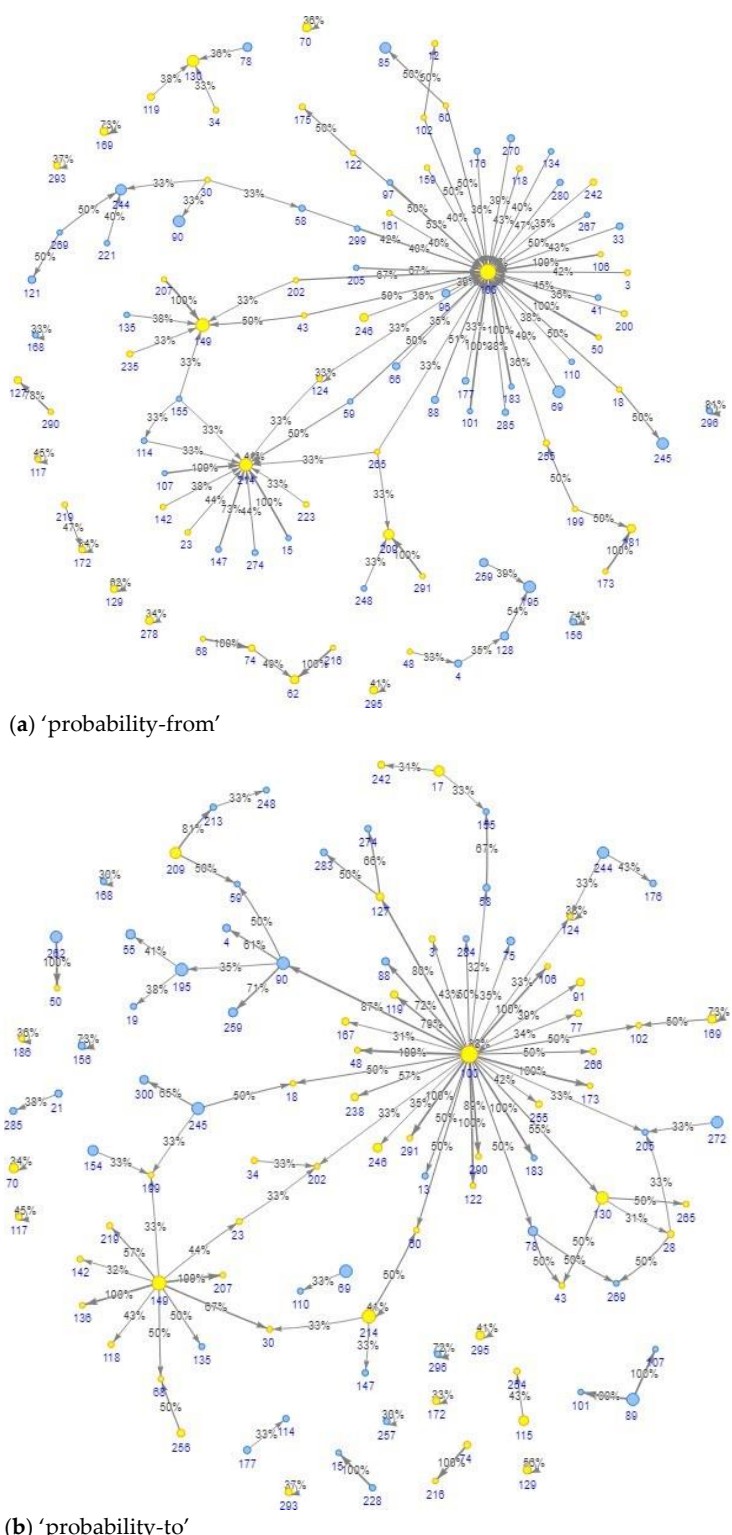

(**a**) 'probability-from'

(**b**) 'probability-to'

**Figure 7.** Cohort A with 'probability-from' and 'probability-to' edge weights: (**a**) represents cohort A with 'probability-from', and (**b**) represents cohort A with 'probability-to'. The yellow nodes are for MHSU (mental health substance use)-related service classes, while the blue nodes are for non-MHSU or medical/surgical service classes.

In the case of 'probability-from', having chosen a node, all arrows from that node are considered. In the case of 'probability-to', having chosen a node, all arrows to that node are considered instead. For example, in Figure 7a, for many nodes, the most probable 'next' node is 100, while in Figure 7b, for many nodes, the most probable 'previous' node is 100. This reflects the fact that 100 is the most used service class for this cohort. However, it can also be noticed that some nodes are more attached to 214 or 149 instead of 100. Note that these plots with a probabilistic edge weight hide the actual number of transitions.

## 5. Discussion

This paper has demonstrated that the dynamics of a patient journey at the local level can be characterized via a next-service-event model. Following the work that was performed in [8,9], this paper injected the order of events into the model to provide a topological depiction of the service system as a whole, reflecting the dynamics of patients' movement across a complex health service system. As well, the strength of coupling between services can be assessed and measured using metrics such as 'probability-to' and 'probability-from'.

As was stated in [8,9], this exercise still relies on the engagement of SMEs and/or SSOEs to select the most appropriate cohorts of interest and determine the appropriate resolution (i.e., the number of edges) of the visualization that emerges from the graph topology. This is due to their understanding of factors that affect service access, such as waiting lists, easiness of accessibility to a service, and service functional integration or lack of it.

Moreover, similar to the manipulation of a macroscope, different cohorts may require different resolutions, depending on: (1) the size of the cohort, (2) the total numbers of interactions between the cohort members and the service system, and (3) the chosen unit of measurement for the weight—number of interactions vs number of patients vs probability of transition. Additionally, the level of homogeneity of patients in a cohort can influence the choice of the resolution. If they share some core features but are highly variable on others, there will be many edges that have low weights, and these will need to be filtered out. In some limit cases where everyone was exactly the same and used the same services in the same order, the filtering will have very little impact on the shape of the topology.

The results show that both Cohort A and Cohort C demonstrate a strong and effective service interconnectivity. For Cohort A, a strong interconnection among some MHSU services can be seen, whereas Cohort C shows an interconnection among certain medical/surgical services. Both of these cohorts demonstrate the evidence of an overarching service delivery model that wraps the services around the patients. This is not the case for Cohort B, where many pathways that move 'from' and 'to' service class 100 can be seen, with very little interconnection among themselves. Cohort B interactions represent a hub and spoke model. Considering that service class 100 represents the Emergency Department, the hub and spoke model shows how dependent Cohort B is on the emergency service to access other required services.

The hub and spoke topologies may be optimal from the standpoint of the efficient use of resources—it requires fewer edges to enable access to all points connected to the hub. However, the distance traveled may be greater than is the case in a point-to-point topology. If distance and time are correlated, and if timely access to services is important, as is the case for healthcare, the hub and spoke model may not be effective. As well, the hub may not be equipped to facilitate transport to all important nodes on the rim. Moreover, a question may arise on why is a service, such as 100, located at the center. Is it a reflection of the easy accessibility to that central service? Is it a reflection of the service range provided at the service location? These two factors, namely easy accessibility and the service range provided, can explain the emergence of 100 as a hub and the subsequent loops. In the case of an emergency, represented in this paper as service 100, it is the hub for reasons of "least action / easy access", not "best access to a full range of services". The emergence of such a topology in a system may be indicative of problems. This may result in the overcrowding

of the emergency departments, a reality that many healthcare organizations around the world are facing [26,27].

These cohort comparison visualizations show how services interoperate for clinical problems that are covered by explicitly articulated clinical practice guidelines. The notion of the "least action", as previously stated, influences the dynamics of patients' engagement with a complex service system. These guidelines introduce the notion of injecting some "force" into influencing the service system and ordering providers to behave in a certain manner. These guidelines are imposed on service system operations by parties directing and delivering care, and are supported by technology. The differences in the visualizations are immediate and striking, as shown in the topological representation of service engagement for Cohort A and Cohort C.

When the quality of care is impacted by the dynamics of patients' movement through the healthcare service system, the methods outlined in this paper may supply information and products to support the efforts in addressing quality issues. They can be very useful tools in the hand of QA/QI to help to inform service delivery changes and optimization initiatives, including the cost–benefit analysis of the various service delivery models.

Integrating service delivery is an important feature of any health service organization. However, it is hard to assess the dynamics of this integration or quantify its effectiveness. Using visualization, the method outlined in this paper can be used to assess the dynamics of an integrated service delivery structure. Additionally, using the concept of graphs and sub-graphs, such a structure's effectiveness can also be quantified.

In a subsequent paper, the findings of this paper will be expanded to analyze the quality of care implication, including treatment response/non-response, for each of the topology. In addition, the dynamic interplay of services and the access (or lack thereof) associated with those dynamics will be explored from a cost and system capacity perspective. Moreover, regarding service delivery integration, the concept of graph and sub-graph will be explored from the perspective of quantifying the effectiveness of the various topologies. Finally, the hub and spoke phenomenon will be explored in more detail to determine to what extent the emergence of such a topology is indicative of problems.

The proximal determinants of health can be distinguished into two broad categories: (1) those that are intrinsic to people and their health conditions, and (2) those that are embodied in the person's interaction with the service system. In the cohort creation and selection phase, these intrinsic person characteristics were considered. However, beyond the consideration of intrinsic person characteristics in the cohort selection, one of the contributions of the methodology proposed in this paper is to provide a way to factor in the dynamic of patients interacting with health services and treat it as a potential proximal determinants of health. However, this can be challenging to identify and measure. This will be explored in more detail in a subsequent study.

The study conducted in this paper tremendously benefited from inputs from team members with a clinical background during the analysis of the cases that were chosen for illustration. For the local implementation of the proposed methodology, it would be ideal to involve patients or persons with lived experience as well at every stage of the analysis. The characteristics of the chosen cohorts of concern drove some of the characteristics of the topologies. The people in these cohorts experience problems directly associated with behaviors that take place within larger social contexts. In order to characterize people in terms of these behaviors, methods such as community engagement or direct observational methods may need to be applied. Also, given the broader social implication, as previously stated, the inclusion of distal determinants of health into the model may be a good extension of the methodology. However, this will require access to a dataset that may not be readily available within a healthcare organization.

This methodology is geared toward identifying the core/most prevalent features, in terms of service engagement, of patients within a cohort. While this represents the usual questions that may need to be answered from an operational perspective, there may be cases where the non-core/least prevalent features may need to be known about patients

within a cohort. When the number of edges is very high, the topological depiction of the cohort does not provide any visibility into the dynamics of engagement with the service for the patients in the cohort. The number of edges needs to be reduced for the resolution to provide a clearer/interpretable image of the topology. For a cohort that shares a core set of features, but that are highly variable in others, by limiting the number of edges, many edges with a low weight will be excluded. If the purpose of the analysis is to know more about the non-core characteristics of the cohort, the methodology outlined in this paper cannot provide an answer. This is a limitation of the proposed methodology.

## 6. Conclusions

This paper has demonstrated that directed graphs can be used to represent a sequence of patient encounter events with the healthcare service system, and the dynamics of patients' journeys with a complex cross-continuum healthcare system can be depicted using various visualizations. Additionally, by carefully selecting various resolutions for the graph topology visualizations, the various characteristics of the patient cohorts can be uncovered, including highly prevalent sequences of patient–service engagement.

The work presented in this paper is the first step in addressing the challenge of locating potential operational problems for patients with complex issues engaging with a complex healthcare service system. Following the analogy of a macroscope, the visualization of the topology, combined with the ability to adjust the resolution of a topological image, by modifying the number of edges in the graph, can help to locate the potential problematic features of the service system. However, more work will need to be conducted in collaboration with various clinical SMEs and SSOEs, incorporating information collected from various clinical guidelines, to expand on the finding of this work to uncover potential operational issues and potential solutions. This will be a focus of future works.

The dynamics of patient engagement will always be driven by multiple factors. These include the healthcare organization's capacity to provide care [28], the capacity of the patients to sustain or not sustain the engagement with the services [29,30], and the structure of the service system [31,32]. The division of the service system into sub-systems that are functionally distinctive is a useful way to provide better care for patients. It is a necessary way of organizing a large and complex dynamic system. Additionally, some sub-systems are more routinely or invariably connected than others. Moreover, patients will always be closer to some services than others, depending on their needs. The goal behind the proposed method is to identify areas of improvement, by providing visibility into the dynamics of patients' engagement and to optimize care for patients with complex problems or chronic conditions that may rely on multiple services that may not necessarily be optimally connected. The methodology is universal, but any implementation of the methodology has to be localized to the reality of the health organization.

**Author Contributions:** Conceptualization, J.B., Y.S., K.M., A.R. and A.K.; data curation, S.R.; formal analysis, J.B., Y.S. and K.M.; investigation, J.B., Y.S. and K.M.; methodology, J.B., Y.S. and K.M.; project administration, J.B.; resources, J.B., K.M. and S.R.; software, Y.S., S.R. and E.C.; supervision, J.B., K.M., A.R. and A.K.; validation, J.B., Y.S., K.M. and E.C.; visualization, J.B., Y.S. and K.M.; writing—original draft, J.B. and Y.S.; writing—review and editing, J.B., Y.S., K.M. and A.R.; funding acquisition, not applicable. All authors have read and agreed to the published version of the manuscript.

**Funding:** This research received no external funding.

**Institutional Review Board Statement:** A certificate of approval was provided by the University of Victoria's Research Ethics Board (REB), following the British Columbia, Canada Ethics harmonization guideline. The REB number is H21-02817.

**Informed Consent Statement:** Not applicable.

**Data Availability Statement:** The datasets presented in this article are unavailable because of privacy or ethical restrictions. Requests to access the datasets require a certificate of approval by the University of Victoria's Research Ethics Board, following the British Columbia, Canada Ethics harmonization guideline.

**Conflicts of Interest:** The authors declare no conflicts of interest.

## Appendix A

A zip file containing the R source code used for graph manipulation and visualization, along with supportive documentation (Readme File), and a sample dataset have been submitted along with this manuscript. Anyone is permitted to use the code. As previously mentioned, the data used for the study will not be made available due to privacy concerns. The sample dataset provided, as part of the appendix, is for illustrative purposes only. However, the methodology described in the manuscript, combined with the source code, can be applied to any dataset similar to the one used for the study.

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
