# Peer review of "Analyzing Patterns of Service Utilization Using Graph Topology to Understand the Dynamic of the Engagement of Patients with Complex Problems with Health Services"

_biomedinformatics, doi:10.3390/biomedinformatics4020060_

Round 1

Reviewer 1 Report

Comments and Suggestions for Authors

    • Dear authors, I appreciate your comprehensive blueprint for the graphic analysis of complex health-related cohorts in a clear and well-articulated writing. However, I have two suggestions that may enhance the impact of your publication.
      1.  As this work serves as an introduction and inspiration in the field, it would be beneficial to include a separate subsection detailing the tool and algorithm used for generating the graphic visualization, and corresponding settings for comparisons in the article, if the corresponding part will not be open-sourced. Beyond the presentation of results, this addition would provide readers with a deeper understanding of your methodology, and particularly making it easier for them to follow and replicate your work.
      2. Even though quantitative analysis may not be the central focus of this article, incorporating it could enhance the overall impact. Providing detailed real-world examples, such as measures of accuracy, precision, and p-values, could draw more readers towards methods based on graph topology.

Author Response

  • Response to suggestion 1 regarding the inclusion of a separate subsection detailing the tool and algorithms used for generating the graphic visualizations, and corresponding settings for comparison
    • Access to Source Code: following the above suggestion, an appendix (page 12) has been added to the manuscript providing details on how to access the R source code used for graph manipulation and visualization, with supportive documentation (Readme File), and a sample dataset, to provide the readers with a deeper technical understanding of the methodology, and make it a little easier to replicate the work. As mentioned in the manuscript, the data used for the study will not be made available due to privacy concerns. The sample dataset provided, as part of the appendix, is for illustrative purposes only. However, the methodology described in the manuscript, combined with the source code can be applied to any dataset similar to the one used for the study.
    • Results Replication: given the methodological nature of the study, the approach proposed in this study can be applied to any set of patient cohorts to analyze and compare their dynamics of engagement with any complex cross continuum healthcare system. The choice of cohorts of interest used for this study and the corresponding results were used for the illustration of the methodology proposed in the manuscript.
  • Response to suggestion 2 regarding the inclusion of quantitative analysis such as accuracy, precision and p-values.
    • Quantitative analysis was not the central focus of this article. However, there an article that we just submitted entitled ”Disparities in access to services, as evident in patient journeys: Illustrating a Nuanced Approach in Assessing Healthcare Equity Using Patterns of Service Utilization Within Longitudinal Encounter Data” that addresses the inclusion of quantitative analysis suggestion. This article expands on the use of graphs and the concept of Patterns of Service Utilization (PSUs), and introduces various quantitative measures such as z-value, in analyzing and comparing various cohorts' access to various healthcare services.

Reviewer 2 Report

Comments and Suggestions for Authors

This study explores the intricate relationship between healthcare services and chronic condition patients, focusing on the role of health determinants, treatment responses, comorbidities, and healthcare service abilities. It advances from prior research that used atemporal models by incorporating time, thus providing a dynamic view of patient navigation through healthcare systems. The paper needs to outline the implications of this temporal perspective, describe the methodologies for data collection, and the place of machine learning in analyzing these patterns. Additionally, it must address practical applications for healthcare providers, ethical considerations, and suggestions for future research enhancements

1.          Can you clarify the proximal determinants of health considered in this study?

2.          How does the treatment response factor into the interaction dynamics?

3.          What specific chronic health conditions were analyzed in this study?

4.          How were comorbidities identified and factored into the analysis?

5.          What indicators measure service system capacity within the study?

6.          Can you detail the previous work that utilized Graph Machine Learning and NLP methods?

7.          What were the limitations of the atemporal data models previously used?

8.          How did reintroducing time into the data model change the analysis outcomes?

9.          What types of topological depictions were generated for the dynamics of patient movement?

10.      How does the time factor influence the interpretation of a complex healthcare system?

11.      Were there any challenges in transitioning from atemporal models to temporal ones?

12.      How does the new temporal model account for seasonal variations in healthcare service demand?

13.      What metrics were used to quantify the dynamics of patient movement?

14.      How was patient movement data collected and processed for this study?

15.      Can you provide examples of insights gained from the topological depictions?

16.      How did you ensure the temporal data model's accuracy and reliability?

17.      What role did machine learning play in analyzing the temporal patterns?

18.      How do you envision healthcare providers applying these topological depictions in practice?

19.      Were there any ethical considerations when modeling patient movement across the healthcare system?

20.      Could you suggest potential improvements or future extensions for this research methodology?

Comments on the Quality of English Language

n/a

Round 2

Reviewer 1 Report

Comments and Suggestions for Authors

I am glad to see the update for both reviewers. Good luck with your publication, authors.